# The Immune Response to SARS-CoV-2 and Variants of Concern

**DOI:** 10.3390/v13101911

**Published:** 2021-09-23

**Authors:** Elham Torbati, Kurt L. Krause, James E. Ussher

**Affiliations:** 1Department of Microbiology and Immunology, University of Otago, Dunedin 9016, New Zealand; ellie.torbati@otago.ac.nz; 2Vaccine Alliance Aotearoa New Zealand, Malaghan Institute of Medical Research, Wellington 6242, New Zealand; 3Department of Biochemistry, University of Otago, Dunedin 9016, New Zealand; kurt.krause@otago.ac.nz; 4Maurice Wilkins Centre for Molecular Biodiscovery, University of Auckland, Auckland 1142, New Zealand

**Keywords:** SARS-CoV-2, COVID-19, variants of concern, immune response, pandemic, vaccine

## Abstract

At the end of 2019 a newly emerged betacoronavirus, severe acute respiratory syndrome coronavirus 2 (SARS-CoV-2), was identified as the cause of an outbreak of severe pneumonia, subsequently termed COVID-19, in a number of patients in Wuhan, China. Subsequently, SARS-CoV-2 rapidly spread globally, resulting in a pandemic that has to date infected over 200 million individuals and resulted in more than 4.3 million deaths. While SARS-CoV-2 results in severe disease in 13.8%, with increasing frequency of severe disease with age, over 80% of infections are asymptomatic or mild. The immune response is an important determinant of outcome following SARS-CoV-2 infection. While B cell and T cell responses are associated with control of infection and protection against subsequent challenge with SARS-CoV-2, failure to control viral replication and the resulting hyperinflammation are associated with severe COVID-19. Towards the end of 2020, several variants of concern emerged that demonstrate increased transmissibility and/or evasion of immune responses from prior SARS-CoV-2 infection. This article reviews what is known about the humoral and cellular immune responses to SARS-CoV-2 and how mutation and structural/functional changes in the emerging variants of concern impact upon the immune protection from prior infection or vaccination.

## 1. Background about COVID-19

In late December 2019, a group of patients with pneumonia-like symptoms of unknown origin were reported from Wuhan, China. Some of them showed additional respiratory symptoms such as acute respiratory distress syndrome (ARDS) and/or severe acute respiratory failure [1,2,3]. Soon after, the disease spread from Wuhan to other cities in China and then to more than 100 countries globally [4]. In March 2020, it was announced as a pandemic by the World Health Organization (WHO) [5]. The etiologic agent of the disease, termed COVID-19 by the WHO, was identified as a newly emerged coronavirus, Severe Acute Respiratory Syndrome Coronavirus 2 (SARS-CoV-2) [3]. 

SARS-CoV-2 belongs to the Coronaviridae family, from which four members (229E, HKU1, NL63 and OC43) are known to cause respiratory infections, most commonly the common cold, in humans. Since 2002, three zoonotic coronaviruses have emerged and caused large epidemics in humans: SARS-CoV in 2002–2004; Middle East Respiratory Syndrome Coronavirus (MERS-CoV) since 2012; and SARS-CoV-2 since 2019. These zoonotic coronaviruses can all cause severe acute respiratory infections and nosocomial outbreaks [6,7].

## 2. Genome of SARS-CoV-2

The SARS-CoV-2 genome is comprised of 14 open reading frames (ORFs) (Figure 1). ORF1a and ORF 1b encode 16 non-structural proteins (known as NSP1–16). These NSPs comprise the replication–transcription complex (RTC), which includes a Papain-like protease (PLP), 3CL-protease, RNA-dependent RNA polymerase (RdRp), and RNA helicase. The remaining ORFs encode accessory proteins and four structural proteins: spike (S), envelope (E), membrane (M), and nucleocapsid (N). Spike mediates SARS-CoV-2 entry into host cells (Figure 1) [8]. Interestingly, the SARS-CoV-2 S gene shares only around 75% sequence identity with the SARS-CoV S gene [9], and these differences, in part, have resulted in the creation of an extended structural loop containing basic amino acids in SARS-CoV-2, which has been implicated in virus transmissibility [9].

## 3. Entry of Virus into Host Cells

The entry of the virus into host cells is mediated through the interaction between the spike glycoprotein and the host receptors (ACE-2 and CD147); human proteases act as entry activators [10,11]. The spike protein functions as a trimer [12]. The spike protein is comprised of 2 main regions, known as S1 and S2. The binding of spike to its receptor is through the S1 region, which is comprised of 2 domains: the N-terminal domain (NTD) and the receptor binding domain (RBD). The RBD constantly changes between an extended conformation, which is utilised for receptor binding, and a drawn-in or contracted conformation for immune evasion [10]. The S2 region (stalk) mediates viral fusion with the host cell membrane and subsequent viral entry [11]. Upon binding of RBD to its receptor, the spike protein is proteolytically activated at the S1/S2 junction, commonly by TMPRSS2, a cell surface protease, but this cleavage can also take place by other proteases including furin and cathepsins, lysosomal proteases. The activation of the spike protein results in S1 dissociation and in structural changes in S2, ultimately leading to fusion of the viral and host cell membranes [10]

## 4. Adaptive Immune Response: Protection vs. Severe Disease

Following SARS-CoV-2 infection, both innate and adaptive immune systems of the host respond to the infection. The adaptive immune responses are antigen specific but are slower than innate responses [13,14]. The adaptive immune response is important in viral clearance and long-lasting immunological memory to protect the host from SARS-CoV-2 reinfection [15].

One of the cardinal features of the adaptive immune response is the rapid clonal expansion of antigen-specific T and B lymphocytes. The time needed to produce an adequate number of adaptive immune cells to control SARS-CoV-2 infection is ~6–10 days after priming [16]. This can vary among people with different severities of COVID-19. The viral load at different phases of infection and the level of immune responses are the main indicators of COVID-19 symptom severity [16]. Severe COVID-19 and excessive immune responses are associated with high and persistent viral loads. Failure to control viral replication early in infection can result in inflammation that can lead to more severe disease and organ damage [17,18].

Antibodies provide protection against SARS-CoV-2 infection, although a humoral immune response is not essential for the control of established infection, as demonstrated in patients with agammaglobulinemia [19]. After infection, SARS-CoV-2-specific B cells differentiate to plasma cells, which produce antibodies specific to the viral antigens. An increased number of B cells have been reported during the recovery phase of COVID-19 [20]. Neutralizing antibodies play an important protective role by blocking the virus from entering the host cells, which limits infection [13]. 

Neutralizing antibodies are specific to viral epitopes that are predominantly in the spike protein [21]. More than 90% of COVID-19 patients have detectable neutralizing antibodies targeting the RBD of the spike protein [16]. Several distinct neutralising epitopes have been defined in the RBD. Yuan et al. classified the antibody epitopes on the RBD into six classes: four RBD subsites (A–D), CR3022, and S309 sites [22]. Structural analysis showed that a large area of the tested antibodies contacted K417, E484 and N501. Most of subsite A binding antibodies mainly covered and interacted with the K417 and N501. All of the IGHV1-2 antibodies (one of the two most frequently elicited antibody families, which can bind to RBD in different modes) bound to the RBS-B epitope and mainly interacted with K417, E484 or N501. The antibodies contacting subsite C mainly interacted with E484 and L452, while the subsite D binding antibodies and CR3022 were not actively bound to any of these residues [22]. In a study by Deshpande, et al., the anti RBD neutralising antibodies were classified into four structural groups (C1–C4) based on their binding epitopes. The epitope targeted by C1 antibodies was located on the RBD ACE-2 binding site. C2 binding antibodies blocked the ACE-2 binding site in both the up and down RBD conformations. C3 neutralising antibodies bound mainly to the outside of the ACE-2 binding site in both conformations. The C4 neutralising antibodies bound to an epitope farther from the ACE-2 binding site, which is only accessible following major conformational change in this area [23]. These findings are in alignment with those of Barnes et al. who also reported 4 main classes of neutralising antibodies [24]. The first class blocked the ACE-2 binding site on RBD in its extended position. The second class of antibodies bound to the RBD in both up and down conformations and bound to residues adjacent to the RBD site. The third class of neutralising antibodies did not bind to the ACE-2 binding site found on the RBD. These antibodies recognised the RBD in both up and down conformations. The fourth class of antibodies bound outside of the ACE-2 binding site but in an up conformation [24].

SARS-CoV-2-specific T cell responses have been detected in almost all COVID-19 patients, with more prominent CD4+ T cell responses compared to CD8+ T cell responses [16,25]. Virus-specific CD4+ T cells may differentiate into Th1 cells, which produce antiviral cytokines such as interferon-gamma (IFNγ), and T follicular helper cells (Tfh). Tfh cells play a crucial role in the development of long-term humoral immunity through the germinal centre reaction, which results in high affinity antibody responses and long-lived plasma cells and memory B cells [16,26]. Grifoni et al. reported that SARS-CoV-2-specific CD4+ T cell responses were predominantly against S, with lesser responses against M and then N proteins [25]. However, in other studies, the CD4+ responses were predominantly directed against the M protein, followed by S and in lesser degrees to the N protein [15,16]. M and S reactive CD4+CD154+ T cells were polyfunctional, producing more than one cytokine (including IFN-γ, IL-2, and TNF-α), compared to the N reactive CD4+ CD154+ T cells [15,27]. Of note, cross-reactive CD4+ T cell responses, both to spike and non-spike peptides, have been detected in stored PBMC samples from prior to the pandemic [25]. Prior exposure to human seasonal coronaviruses (NL63, 229E, HKU1 and OC43) could generate SARS-CoV-2 cross-reactive T cell responses [28].

CD8+ T cells also play an important role in the clearance of SARS-CoV-2 infection by secreting cytokines and killing infected cells. SARS-CoV-2 CD8+ T cells specific for a range of SARS-CoV-2 antigens, such as S, N, M, and ORF3a, have been identified [16,25]. High expression levels of IFNγ, granzyme B, perforin, and the marker of degranulation, CD107a, by CD8+ T cells were reported in acute phases of COVID-19 [16,29]. 

In the study by Mazzoni et al., a significant correlation between the levels of SARS-CoV-2-specific IgA, IgM, and IgG antibodies and the frequency of virus-reactive CD4+CD154+ T cells was reported [15]. The correlation between the antibodies and antiviral CD4+ have also been reported in other studies [25,30,31].

## 5. Adaptive Immune Responses in Asymptomatic, Mild, and Severe Cases of COVID-19

COVID-19 displays a wide spectrum of clinical presentations ranging from mild to severe. If the early host immune response cannot clear and control the virus, the disease will progress toward a secondary phase, which is characterized by an uncontrolled host inflammatory response. This can result in end organ damage [15]. Antigen-specific adaptive immune responses have been reported in both asymptomatic and symptomatic SARS-CoV-2 patients, with a positive correlation between the humoral immune response, and in some but not all studies T cell immune memory, and disease severity [16].

Antibodies against SARS-CoV-2 are detectable in almost all patients after infection. A study by Liu, et al. on 52 convalescent patients over a course of 6 months post symptom onset, reported that anti-S IgG remained detectable in more than 90% of patients [32]. In another study by Whitcombe et al., convalescent sera from patients with mild to moderate COVID-19 were collected up to 8 months post symptom onset [33]. It also showed that after 4 to 8 months 99% of the tested sera contained IgG antibodies against RBD and that 96% showed detectable titres of anti-S IgG [33]. Lee et al. reported a significant increase in neutralising antibody levels which reached its peak 31–35 days post symptom onset [34]. 

The humoral immune response differs with disease severity. While patients with mild or moderate symptoms had an increased frequency of circulating Tfh CD4+ T cells and germinal centre B cells, in patients with severe disease there was a profound reduction of circulating CD4+ T cells and B cell lymphopenia [14]. Consistent with this, Peng et al. reported a negative correlation between the size of the total circulating CD4+ T population and the COVID-19 severity [35]. Despite this, higher levels of B cell receptor clonal expansion and B cell activation have been detected in patients with severe COVID-19, indicating that there is a robust humoral immune response in patients with severe infection [1]. Plasma from people previously infected with SARS-CoV-2 was shown to contain higher titres of antibodies against SARS-CoV-2 after severe infection [36]. Significantly higher levels of antibodies specific for spike, RBD, and N proteins were reported in severe cases compared to the mild cases of COVID-19 [35]. Similarly, Zhang et al. found that the levels of anti-SARS-CoV-2 antibodies were higher in severe cases compared to mild cases and that the ratio of IgA and IgG to IgD  +  IgM positively correlated with disease severity [27]. 

SARS-CoV-2-specific T cells appear to play a protective role against severe disease. In mild disease, rapid induction of SARS-CoV-2-specific CD4+ T cells was seen and was associated with accelerated viral clearance [37]. Strong memory T-cell responses were seen in convalescent patients [38]. While Peng et al. reported higher overall SARS-CoV-2-specific T cell responses in severe cases compared to patients with mild COVID-19, in patients with mild disease, a larger proportion of the T cell responses to spike protein and M and N were contributed by CD8+ T cells compared with those patients with severe COVID-19 [35]. In severe or fatal COVID-19 patients more than 22 days post-symptom onset, an extended absence of SARS-CoV-2-specific CD4+ T cells was reported [39,40]. In contrast, Mazzoni et al. showed a reduced frequency of polyfunctional M-reactive CD4+ T cells in asymptomatic patients compared with symptomatic COVID-19 [15]. 

## 6. Recent Emergence of Variants and the Theories around the Emergence of New Variants 

The SARS-CoV-2 virus has been notable for the large number and variety of genomic variations recorded since the onset of the pandemic [41]. This variation has produced a large number of changes in all the structural proteins which, through natural selection, has resulted in the production of variants with improved transmission and replication ability [42]. In the case of coronavirus, the major cause of its extensive variability is thought to be due to its notable potential for recombination [43]. Replication errors occur that stem from the coronavirus RNA dependent RNA polymerase but these are reduced, when compared to other RNA viruses, because of its proofreading capability [43].

Since the emergence of SARS-CoV-2 in December 2019 till July 2021, 3945 variants have been reported in Nextstrain. Based on this data, the predicted mutation rate is 25.048 substitutions/year (Figure 2). The number of mutations varies among the different reported variants. As of this writing, the largest number of mutations (*n* = 57) has been reported in variant USA/LA-EVTL2800/2021, which was isolated from a patient in the USA. This is a variant of B.1.1.7 strain and most of the new mutations in this variant occurred in the S gene. 

During uncontrolled viral replication, as found in a large pandemic, viral variants can emerge if they provide either increased infectivity, immune escape, or both. The variants of concern (VOCs) of SARS-CoV-2 described to date contain clustering of non-synonymous mutations in the S gene and have displayed both of these features.

Some aspects of the theory behind the rapid emergence of new variants are controversial. The remarkably high number of mutations in the S gene, the array of mutations in the non-Spike genes, and the high sequence coverage suggest that the new variants have not emerged through gradual accumulation of mutations. It is also unlikely that vaccination has yet exerted sufficient selection pressure to explain the emergence of these variants, although is likely to be a significant factor as vaccine coverage increases. One possible explanation for the emergence of variants is rare prolonged SARS-CoV-2 infection in individual patients who are immunocompromised. It is postulated that a prolonged infection in an immunocompromised host would offer a greater opportunity for mutations and recombination to develop under immune selection pressure through multiple viral replication cycles [44]. Previous studies have reported prolonged infection with SARS-CoV-2 in immunocompromised patients despite receiving convalescent plasma treatment [45,46]. For instance, a 45-year-old immunocompromised patient with a SARS-CoV-2 infection lasting over 5 months has been reported. Genetic analysis of the SARS-CoV-2 samples isolated from this patient over the course of 152 days showed rapid viral evolution, predominantly in the S gene (with 13% of these mutations leading to amino acid changes in the S1 protein) [47,48]. In another immunosuppressed patient, persistent SARS-CoV-2 infection over 4 months was demonstrated [49]. Sequencing of multiple samples over time showed that 18 mutations in the S gene accumulated at a rate of 1.67E-3 mutations/nucleotide/year, which is higher than the average rate of SARS-CoV-2 evolution; initial evolutionary studies reported a mutation rate of ~1E-3 mutations/nucleotide/year [49,50]. The reported mutations in this case caused changes in the spike protein, including in the neutralising antibody epitopes in RBD. The abundance of nonsynonymous mutations in new SARS-CoV-2 variants may also reflect escape from CD8+ T cell epitopes [49].

Another factor that facilities the rapid genetic divergence in SARS-CoV-2 is recombination, which occurs at higher frequency in positive sense RNA viruses, including SARS-CoV-2 [51,52]. The S gene of coronaviruses has been reported as a recombination hot spot [53]. The putative recombination region is detected in the RBD of S protein [54]. Recombination could happen in a new host or in the same host with prolonged infection or could happen when there is a co-infection with different SARS-CoV-2 variants, which could lead to the emergence of more virulent variants [55].

## 7. SARS-CoV-2 Variants of Concern (VOC)

A variant that shows a higher rate of transmissibility, more severe disease followed by higher rate of hospitalisations or death, a significant reduction in neutralisation by antibodies, reduced effectiveness of treatments or vaccines, or failure to be detected in diagnostics assays is considered to be a variant of concern (CDC: https://www.cdc.gov/coronavirus/2019-ncov/variants/variant-info.html, accessed on 8 September 2021, Concern and WHO: https://www.who.int/en/activities/tracking-SARS-CoV-2-variants/, accessed on 8 September 2021). These variants contain important mutations and are classified generally by those changes that are located within ORF1a, 1b and the S protein [56]. Four notable variants of concern have been described to date, including the alpha variant (also known as B.1.1.7, 20I/501Y.V1, VOC 202012/01, or the UK variant), the beta variant (also known as B.1.351, 20H/501Y.V2, or the South African variant), the gamma variant (also known as P.1, GR/501Y.V3 or the Brazilian variant), and the delta variant (also known as B.1.671.2, G/478K.V1 or the Indian variant (Table 1 and Table 2). 

The alpha variant was first isolated in December 2020 in the United Kingdom. It has been reported that the transmission rate of this variant is 71% higher than the ancestral strain originally isolated in Wuhan (Kidd et al., 2021). In addition, there is a significantly higher viral load in infections with this variant [89]. Distributed through the ORF1a, ORF1b and S genes, the alpha variant carries 14 non-synonymous mutations and three deletions, including N501Y which changes the confirmation of RBD. Another significant mutation in this variant is the deletion at position 69/70 of spike (Figure 3 and Figure 4). This mutation results in false negative results in a diagnostic RT-PCR assay targeting the S gene [90].

Around October 2020, after the second wave of COVID-19, a new variant of SARS-CoV-2 was reported in major metropolitan areas in South Africa. This new variant, named as the beta variant or B.1.351, has multiple mutations in the S gene resulting in amino acid changes in spike protein. The beta variant showed three mutations in RBD (K417, E484K and N501) (Figure 3 and Figure 4) [76]. This new variant has been reported in 113 countries and is important due to the ability of the virus to escape neutralising antibodies [76].

The gamma (P.1) variant was first detected in travelers from Brazil who arrived in Japan in early 2021. The variant carries mutations that result in 17 unique mutations including: three deletions, four synonymous mutations, and a 4-nucleotide insertion. This variant contains three mutations in RBD of the spike protein: K417T, E484K, and N501Y [91] (Figure 3 and Figure 4).

In May 2021 the delta variant (B.1.617.2) emerged in India, where it is now the most frequent strain, with a reported 97% increase in transmissibility (Table 2). The delta variant spike carries seven nucleotide changes and deletions compared to the spike of the ancestral Wuhan variant. Two of these mutations are in the NTD motif of the spike protein (T19R and del 157-8), two in the RBD (L452R and T478k), and three more outside these regions (D614G, P681R and D950N). One mutation is located close to the furin cleavage site (P681R) and the D950N mutation is in the S2 motif of the protein [64] (Figure 3 and Figure 4).

## 8. The Effect of the Mutations on the Structure and Function of the SARS-CoV-2 Spike Protein

The evolution of SARS-CoV-2 is moulded by functional constraints and pressure to evade the immune response. The mutations, especially in the VOCs, play important roles in transmissibility and viral escape from neutralising antibodies. Amongst all the mutations, those on spike protein play a particularly crucial role due to the spike protein’s role in virus entry to the host cell and because it is the target of most protective antibody responses.

Generally, the mutations on the spike protein can be classified in three major classes. The first class of mutations is in the RBD [92]. These mutations can result in immune escape from neutralising antibodies and changes in viral fitness. Mutations in RBD can reduce the neutralisation potency of sera from both vaccinated and naturally infected individuals [86,93]. As discussed above, most neutralising anti-SARS-CoV-2 antibodies target the RBD of the spike protein, and escape mutations have generally been reported in the antibody-RBD interface. A study by Starr et al., mapped the mutations that allow the virus to escape from monoclonal antibodies used to treat COVID-19 [94]. Interestingly, not all the mutations located on the antibody contact residues cause viral escape. Conversely, several mutations of residues not in contact with neutralising antibodies (e.g., E406W) have been reported to be important in immune evasion of the virus [94]. The E484K mutation, which has been reported in beta and gamma variants and has now been identified in a new alpha variant (B.1.1.7 + E484K variant), is an escape mutation which requires higher titres of antibodies to neutralise the virus in vitro [95]. E484K has been linked to cases of reinfection in Brazilian patients who had previously been infected with B.1.1.33 variant (E484) and were reinfected with variant B.1.1.28, which contains the E484K mutation [96,97]. Another important RBD mutation is the N501Y mutation that has been reported in alpha, beta, gamma, and other variants such as theta (P.3), B.1.x, B1.621 and A.27 variants. The N501Y mutation, also known as a mutation of major concern, has not been found in the delta variant [98,99]. The 501 residue is one of six RBD contact residues with the ACE-2 receptor. The N501Y mutation permits the variant to be more transmissible, by increasing the affinity of spike protein for its receptor, ACE2 [62,100]. The N501Y mutation improves the viral fitness in the upper airway, likely by enabling a Pi-Pi interaction between spike 501Y and ACE2 41Y residues [101].

The second class of mutations is located in the NTD. There is evidence for immune selection in this region and preliminary evidence that at least one of these changes, delH69/delV70, could improve the viral fitness [102]. The mutations in the NTD are also important because of the presence of a site that is recognised by all NTD-specific neutralising antibodies [103]. This site is called the “supersite” and comprises of three regions: spike residues 14–20, 140–158 and 245–264 and it is both glycan free and electropositive [103,104]. Mutations in this site have been reported in all VOC. The R246A mutation in the NTD reduces binding by monoclonal antibodies targeting the NTD [103]. In addition, the H146Y mutation in NTD was associated with reduced antibody detection by the majority of antibodies tested by McCallum et al. However neither the 69/70 deletion nor the A222V substitution affect the efficiency of antibody binding [103]. In the study by Cerutti, et al., the escape of the alpha variant from NTD-specific antibodies was shown to be due to the deletions of 69/70 and 144/145. The escape of the beta variant from NTD-specific antibodies was due to the deletion 242/244 and the R246I mutation [104].

The third class of mutations are those near the furin cleavage site (FCS). The presence of FCS, which is absent in the spike proteins of other lineages of β-coronaviruses, including SARS-CoV, contributes to SARS-CoV-2′s high infectivity and transmissibility [105]. Mutations near the FCS influence the refolding of the 6-helix bundle [101]. An example of this class of mutation is P681H/R (presents in alpha and delta variants), which is immediately adjacent to the FCS. This mutation provides a better conformation for hydrolysis by TMPRSS2 (a serine protease) and thus augments viral fusion with the host cell membrane [98]. SARS-CoV-2 entry to the target cells is dependent upon both ACE-2 and TMPRSS2. The virus engages ACE-2 as the entry receptor and employs the cellular serine protease TMPRSS2 for S protein priming [85].

In contrast to B cell epitopes, T cell responses to SARS-CoV-2 appear to be relatively intact [86]. While some T cell epitopes may be lost [86], the breadth of the T cell responses ensures ongoing T cell recognition.

## 9. Efficacy of Immunity from Prior Infection against SARS-CoV-2 Variants of Concern

There are few little data about the effect that the VOCs have on the risk of reinfection with SARS-CoV-2. Immune responses following infection with SARS-CoV-2 provide protection against reinfection in the majority of patients for months [106]. The risk of reinfection is reduced by 83% for at least 5 months [107]. While VOCs carry mutations which elude neutralisation by the antibodies and there are reports of reinfections with the VOCs, it is unclear whether the risk of reinfection is increased. A study by the Public Health England reported that 0.06% of cases (44 out of 6614) represented probable reinfections over a time period of 5 months (June–November 2020) [106,108]. In a study from the UK from September 2020 to December 2020, 0.7% of tested individuals became reinfected. However, the rate of reinfection in different regions did not correlate with the proportion of infections in those regions caused by the alpha variant, suggesting the alpha variant does not have a major effect on the risk of reinfection [109]. Another example is the emergence of the gamma variant in Manaus, Brazil, in a population with high rates of prior SARS-CoV-2 infection based on seroprevalence studies [110]. A longitudinal serological study of unvaccinated repeat blood donors in Manaus showed that following the emergence of the gamma variant 16.9% of presumed infections had serological evidence of previous infection with SARS-CoV-2 [111].

## 10. Currently Available Vaccines and Their Efficacy

Despite being little more than 18 months into the pandemic, there are already four vaccines available that have been approved and widely used around the world: Pfizer/BioNtech’s BNT162b2, Moderna’s mRNA-1273, AstraZeneca’s ChAdOx1, and Janssen’s Ad26.COV2S. The Pfizer/BioNtech, Moderna and Janssen vaccines have been given emergency use authorization by the FDA (https://www.cdc.gov/coronavirus/2019-ncov/vaccines/different-vaccines.html, accessed on 10 July 2021) and the AstraZeneca vaccine is widely approved outside of the US. Other vaccines in use include: Gamaleya’s Gam-COVID-Vac (Sputnik V), SinoVac’s CoronaVac, and Sinopharm’s BBIBP-CorV [112]. Noavax’s NVX-CoV237 has completed phase three trials and is awaiting regulatory approval. With the exception of the inactivated vaccines, all the vaccines utilise spike as the antigen.

Different vaccines have shown varying efficacy in clinical trials. CoronaVac and BBIBP-CorV are SARS-CoV-2 inactivated vaccines and it is reported that CoronaVac has an efficacy of 83.5% [113] and Sinopharm 73–78% 14 days post vaccination [114]. Among the viral vectored vaccines, Gam-COVID-Vac (Sputnik V) has an efficacy rate of ~91% seven days after the second dose [112,115], Ad26.COV2S, which is currently a single dose vaccine, has an efficacy of 66.9% in preventing moderate to severe COVID-19 2 weeks post-vaccination [116], and ChAdOx1 showed 70.4% efficacy 14 days after the second dose [117].

The mRNA vaccines BNT162b2 and mRNA-1273 have been shown to be highly efficacious. BNT162b2 vaccine showed 95% efficacy from 7 days after the second dose [118]. This was confirmed in a post-implementation study in Israel where the effectiveness of the vaccine was reported to be 92% after seven days [119]. In a study by Sahin et al., the BNT162b2 vaccine induced neutralising antibodies and poly specific CD4+ T cell responses in 100% of tested individuals and CD8+ T cell responses in 90% of tested individuals [120]. mRNA-1273 showed 94% efficacy in preventing COVID-19 14 days post vaccination [121,122]. The efficacy of the vaccine after first dose was 80% [123]. mRNA-1273 vaccine induced both humoral and cellular anti-SARS-CoV-2 immune responses, with the titre of anti-SARS-CoV-2 antibodies induced by the vaccine being much higher than convalescent serum over the course of 100 days [122].

NVX-CoV2373 (Noavax) is currently the only SARS-CoV-2 protein-based vaccine that has completed clinical trials, although it has yet to receive regulatory approval. This vaccine is made of SARS-CoV-2 trimeric spike protein and Matrix-M1 adjuvant. An efficacy greater that 95.6% was reported against the original B1 viral strain. CD4+ responses were also detected in individuals vaccinated with NVX-CoV2373 [124].

## 11. The Efficacy and Effectiveness of the Vaccines against Variants of Concern

While the effectiveness of vaccines at protecting against symptomatic infection with VOCs may be decreased, the data to date suggests that they remain effective at preventing severe disease, hospitalisation, and death. It is currently unclear whether variant-specific boosters will be required in the future.

There are limited data available about the efficacy of the vaccines against VOCs. In a trial in South Africa, the efficacy of ChAdOx1 against the beta variant was very low (10.5–22%) [81]. In a trial in the UK, the efficacy of ChAdOx1 was reported to be higher against the alpha variant at 70.4% [74]. NVX-CoV2373 showed 86% and 60% efficacy against the alpha and beta variants respectively [124]. An efficacy of 51% against the beta variant for NVX-CoV2373 was also reported by Shinde et al. [125]. The efficacy of Ad26.COV2S was 52% at least 14 days post vaccination and 64% at least 28 days post vaccination against moderate to severe COVID-19 in South Africa, where ~94% of cases were infected with the beta variant [116].

Multiple studies have now assessed the effectiveness of vaccines against VOCs. The effectiveness of single dose vaccination against symptomatic infection with the alpha variant was higher in individuals vaccinated with mRNA-1273 (83%) compared to those vaccinated with BNT162b2 (66%) or ChAdOx1 (64%). The effectiveness increased after the second dose of vaccination (effectiveness of 89% for BNT162b2 and 92% for mRNA-1273) [126]. Similarly, in a study by Hall et al., the effectiveness of BNT162b2 against symptomatic infection with the alpha variant was 70% from 21 days after the first dose and 85% from 7 days after the second dose [127]. In another study in Qatar, the effectiveness of BNT162b2 against documented COVID-19 cases with alpha and beta variants was 89.5% and 75%, respectively [128]. Several studies have shown that a single dose of BNT162b2 has reduced effectiveness against alpha, beta, and delta variants [128,129,130]. The effectiveness of a single dose of mRNA-1273 (77%) against symptomatic infection by beta or gamma variants was higher compared to BNT162b2 (60%) and ChAdOx1 (48%) [126]. Amongst health care workers in Manaus, Brazil, who had received at least one dose of CoronaVac, the vaccine was 49.6% effective at preventing symptomatic SARS-CoV-2 infection at a time when the gamma variant was prevalent [131]. A recent study showed that a single dose of BNT162b2 vaccine and ChAdOx1 was only 36% and 30% effective against the delta variant, while with two doses the efficacy increased to 87.9% and 66.1% respectively [129]. In a study in Canada, it was reported that the effectiveness of single dose vaccination with BNT162b2 (56%) or mRNA-1273 (72%) against the delta variant was lower than against the other variants, while the effectiveness of a single dose of ChAdOx1 (67%) against the delta variant was similar to the effectiveness against the alpha variant (64%). In fully vaccinated individuals the effectiveness of BNT162b2 against the delta variant increased to 87%, while the effectiveness against the delta variant of two doses of mRNA-1273 or ChAdOx1 was not reported [126].

Effectiveness against VOCs may wane with time since vaccination. A recent study in Israel reported an increase in breakthrough infections in those who had been vaccinated early (January 2021) with BNT162b2 compared to those who had been vaccinated more recently (April 2021). In June and July 2021, the risk of breakthrough infection in completely vaccinated individuals was increased 2.26 fold in early vaccinees compared to those vaccinated more recently. During the study period, the delta variant was the dominant circulating strain in Israel [132]. On the 30 July 2021, a third dose of BNT162b2 was approved in Israel for people over 60 years old who had been vaccinated at least 5 months previously. In an analysis of data to the 24 August 2021, increased short term protection against confirmed infection and severe disease was reported with the third dose [133]. Further follow-up is required to determine whether a third dose increases vaccine effectiveness long term.

## 12. The Neutralising Activity of Post-Vaccine Sera against VOCs

While only a small decrease in vaccine effectiveness has been reported, due to the presence of mutations in spike in neutralising antibody epitopes, a decrease in the ability of post-vaccine sera to neutralise VOCs in vitro has been seen (Table 2). Tauzin (2021) found no neutralising activity against the beta variant in plasma from SARS-CoV-2 naive individuals who had been vaccinated with BNT162b2 vaccine. In contrast, strong antibody-dependent cellular cytotoxicity (ADCC) was reported. In comparison, previously infected individuals who were subsequently vaccinated showed a significant increase in pre-existing ADCC and neutralising antibodies against the beta variant [134]. A study by Edara et al. on convalescent and post-vaccination (mRNA-1273 or BNT162b2) sera, reported that the majority (79%) of convalescent sera and all post-vaccination sera were able to neutralise the B.1.617.1 variant. However, this neutralising activity was 6.8 fold less than against the ancestral Wuhan strain [88]. Wang et al. reported similar neutralising antibody levels following vaccination (mRNA1273 or BNT162b2) or natural infection. RBD-binding monoclonal antibodies were generated from both vaccinees and convalescent patients; monoclonal antibodies from both groups bound to similar epitopes and most had reduced or no ability to neutralise pseudoviruses with K417N, E484K or N501Y mutations [135]. In another study, sera from individuals vaccinated with BNT162b2 or ChAdOx1 showed a 3- to 5-fold reduction in antibody neutralisation against the delta variant compared to the alpha variant [64]. These findings align with those of Madhi et al., who showed the reduced ability of post-vaccination (ChAdOx1) sera to neutralise the beta variant in both pseudovirus and live virus assays [81]. A study using post-vaccination sera from the phase one trial of mRNA-1273 showed no significant change in neutralising activity against the alpha variant compared with ancestral SARS-CoV-2 in a pseudovirus neutralisation assay [73]. In the same study, a 6.4-fold reduction in the ability of post-vaccination sera to neutralise beta variant was reported [73]. These results align with the study by Shen et al., who studied the neutralising ability of post-vaccination sera of individuals vaccinated with mRNA-1273 or NVX-CoV2373. In a pseudovirus neutralisation assay, they found a 2- and 2.5-fold reduction in the neutralisation of epsilon (B.1.429) variant by sera post vaccination with mRNA-1273 or NVX-CoV2373, respectively, compared with an ancestral variant containing D614G. In contrast, a 9.7- and 14.5-reduction in the neutralisation of the beta variant was reported by sera post vaccination with mRNA-1273 or NVX-CoV2373, respectively [82]. Lustig et al. examined the ability of sera from convalescent COVID-19 patients who had subsequently been given one dose of BNT162b2 to neutralise the original B.1 virus, and the alpha, beta, and gamma variants in live virus neutralisation assays. Compared to pre-vaccination levels, neutralising antibody titres 1–2 weeks post-vaccination were 114, 203, 81, and 228 times higher against B.1 and alpha, gamma, and beta variants respectively [136].

## 13. Heterologous Prime-Boost Vaccination and Variants of Concern

Heterologous prime-boost vaccination could be a strategy to address variants of concern that carry immune escape mutations. Heterologous boosters (using either a different vaccine type, a variant antigen, or both) could potentially increase protection against variants of concern by increasing the magnitude of humoral and cellular immune responses and/or by increasing the breadth of those responses [137,138,139,140]. Booster vaccines could either use the spike protein from the original Wuhan strain of SARS-CoV-2 or the spike protein from a variant of concern. Most research on heterologous prime-boost vaccination strategies is at an early stage. To date, these strategies appear to be safe and immunogenic. However, data on the impact on neutralisation of variants of concern are only available from a few studies.

The ongoing Com-COV study [141] is comparing homologous and heterologous prime-boost vaccination with BNT162b2 and ChAdOx1. When boosted at 28 days, higher levels of antibodies and T cell responses were reported in individuals who were primed with ChAdOx1 and boosted with BNT162b2 compared to those who were primed with BNT162b2 and boosted with ChAdOx1 [141]. The heterologous schedules induced higher levels of antibodies than a homologous prime-boost with ChAdOx1 but not with BNT162b2 [141]. The strongest T cell responses were seen in individuals primed with ChAdOx1 and boosted with BNT162b2 [141]. In the ongoing Com-COV2 study, participants who have been primed with ChAdOx1 or BNT162b2 will be boosted with mRNA-1273 or NVX-CoV2373 [142].

In a Spanish study, adults aged 18 to 60 years were primed with a single dose of ChAdOx1 and boosted after 28 days with BNT162b2 [143]. Fourteen days post boost, the geometric mean titres of anti-RBD IgG, anti-spike IgG, and neutralising antibodies were 77.7-, 36.4-, and 45.6-fold higher, respectively, in those who were boosted with BNT162b2 than in those who were not. Similar boosting of T cell responses was seen 14 days post boost, with a 4.2-fold increase in IFN-γ production in response to pools of peptides from the SARS-CoV-2 spike in those who had received the booster [143].

In healthcare workers in Germany, Hillus et al. compared the immunogenicity of homologous (BNT162b2: 3-week interval; ChAdOx1: 10–12 week interval) and heterologous (ChAdOx1 prime/ BNT162b2 boost: 10–12 week interval) prime-boost regimens [144]. Compared to the groups that received homologous BNT162b2 or ChAdOx1 regimens, the group who received the heterologous boost showed higher humoral (anti-RBD: no difference; anti-S1-IgG avidity: median relative avidity index 93.6% with heterologous boost versus 73.9% with BNT162b2 and 71.7% with ChAdOx1 respectively; surrogate virus neutralisation titre: 97.1% with heterologous boost vs 92.4% for ChAdOx1) and cellular immune responses (IFN-γ production in response to SARS-CoV-2 S1 peptide pools: 2.4-fold and 4.5-fold higher respectively). Importantly, priming with ChAdOx1 and boosting with BNT162b led to increased serum neutralising activity against the alpha (2.6-fold and 4.5-fold higher than BNT162b2 and ChAdOx1 respectively) and beta variants (5.8-fold and 8.6-fold higher than BNT162b2 and ChAdOx1 respectively) [144].

An ongoing phase two study by Moderna is investigating the effect of a booster dose of mRNA-1273, mRNA-1273.351 (which encodes for the S protein of the beta variant), or mRNA-1273.211 (a 1:1 mix of mRNA-1273.351 and mRNA-1273) in individuals who have previously received two doses of mRNA-1273 [145]. Preliminary results for the mRNA-1273 and mRNA-1273.351 boosters demonstrated that two weeks after receiving the booster, neutralising antibody titres against the wild-type (Wuhan with D614G mutation), beta, and gamma variants increased to a similar or higher levels as reported following the primary vaccine series. Individuals who received the heterologous mRNA-1273.351 boost showed increased neutralising antibody titres (ID_50_) against the beta variant (geometric mean titre 1400 vs 864) but not the gamma variant (geometric mean titre 1272 vs 1308) or the wild type (geometric mean titre 3703 vs. 4588), when compared to those who received the homologous mRNA-1273 boost.

The Moderna beta-variant-specific vaccines, mRNA-1273.351 and mRNA-1273.211, have also been tested in mice [146]. Compared with mRNA-1273, a primary two-dose vaccination course with mRNA-1273.351 resulted in a small (1.4-fold) increase in neutralising antibody tires against the beta variant but a 6.1-fold decrease in neutralising antibody titres against the wild-type (Wuhan with D614G mutation), a 2.6-fold decrease against the gamma variant, and a 3.8 fold decrease against B.1.427/B.1.429. In contrast, a primary vaccination course with the mixed vaccine mRNA-1273.211 resulted in relatively preserved neutralising antibody titres against the wild-type (0.9-fold), gamma (1.3-fold), and B.1.427/B.1.429 (0.6-fold) variants and increased titres against the beta variant (2.4 fold) and compared to mRNA-1273. When mice previously vaccinated with two doses of mRNA-1273 were boosted on day 213 with mRNA-1273.351, a 4.5-fold increase in neutralising antibody titres was seen against the wild-type virus and 15-fold against the beta variant [146].

In another preclinical study from China, the immunogenicity of a range of heterologous prime-boost regimens was assessed in mice [147]. Four different types of vaccines (Sinopharm’s BBIBP-CorV inactivated virus vaccine, CanSino’s Ad5-nCoV adenovirus vector vaccine, Anhui Zhufeu Longcom’s ZF2001 recombinant RBD vaccine, and the People’s Liberation Army Academy of Military Sciences/Walvax Biotech’s ARcoVax mRNA vaccine), which were developed and manufactured in China, were tested on BALB/c mice. Their results showed that the order of vaccination is an important determinant of the level of neutralising antibody production. In mice that were primed with inactivated virus (BBIBP-CorV) followed by the adenovirus vector (Ad5-nCoV), a 6.7-fold higher level of anti-S-IgG was seen than in those primed with Ad5-nCoV and boosted with BBIBP-CorV. Both of these schedules showed significantly higher neutralising antibody titres compared to homologous prime-boost vaccination with either BBIBP-CorV or Ad5-nCoV [147]. Comparing to all heterologous vaccine combinations, priming with the adenovirus vector vaccine and boosting with either the inactivated virus, recombinant RBD, or mRNA vaccines resulted in the highest titres of neutralising antibodies. In contrast, the strongest T cells responses (S-antigen-specific IFN-γ production by T cells) were seen in mice primed with recombinant RBD followed by boosting with the adenovirus vector vaccine [147]. Neutralisation of variants of concern was not assessed.

## 14. Conclusions

There is evidence that variants, especially VOCs, have developed various levels of escape from neutralising antibodies through mutation. Nonetheless, cellular immune responses remain largely preserved. There is a lack of clinical data about the risk of reinfection with VOCs. Vaccine effectiveness is generally reasonably well preserved, but two doses may be required. Importantly, vaccines seem to continue to protect against severe disease. Therefore, current vaccines may be sufficient to provide individual protection against severe disease, but there may be a reduction in protection against infection and transmission to others, and hence community protection (herd immunity). It remains to be seen whether the spike protein is able to mutate further to escape immune responses while maintaining high levels of infectivity. Along with the kinetics of immune responses and post vaccine effectiveness, the ability of SARS-CoV-2 to mutate will determine whether updated vaccines or boosters will ultimately be required, and if so at what frequency.

## Figures and Tables

**Figure 1 viruses-13-01911-f001:**
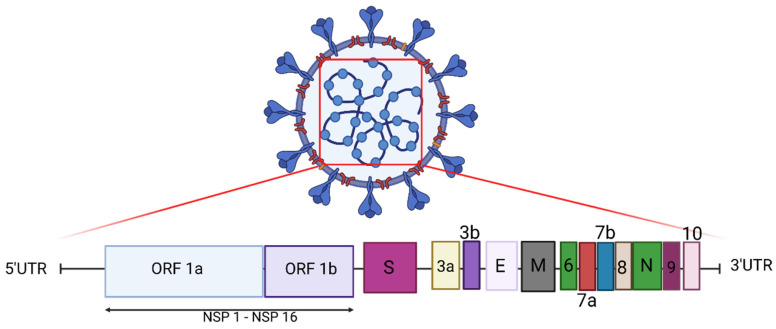
The genomic structure of SARS-CoV-2. The genome encodes two large open reading frames (ORFs), ORF1a and ORF1b, which encode 16 non-structural proteins (NSP1-NSP16). The structural genes encode the structural proteins, spike (S), envelope (E), membrane (M), and nucleocapsid (N), and the accessory genes (3a, 3b, 6, 7a, 7b, 8, 9 and 10) (Created with Biorender.com accessed on 9 September 2021).

**Figure 2 viruses-13-01911-f002:**
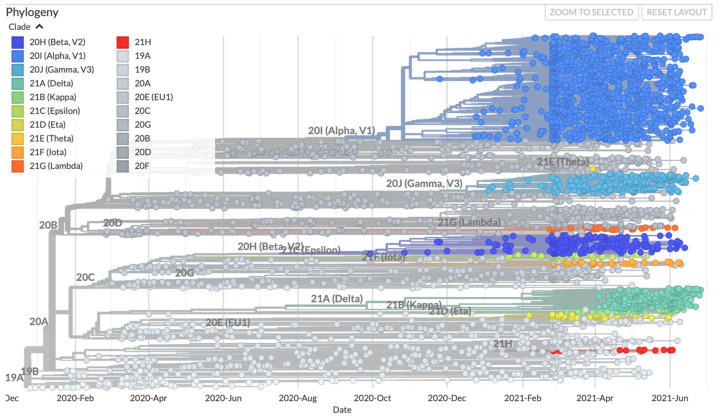
The emergence of new variants over time. The colours represent the ancestral clade. The variants of concern and variants of interest are shown on the graph. The image was taken from www.nextstrain.org under a CC-BY-4.0 license and is unchanged (accessed on 15 July 2021).

**Figure 3 viruses-13-01911-f003:**
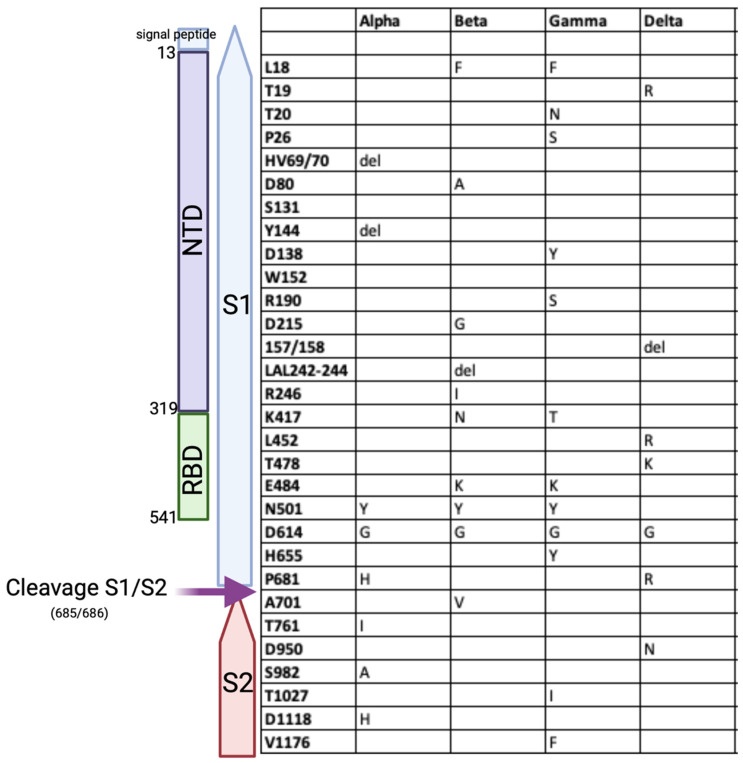
The mutations in the spike protein of four variants of concern. The schematic design of the spike protein is shown on the left. The mutations in 4 variants of concern are shown in the table on the right.

**Figure 4 viruses-13-01911-f004:**
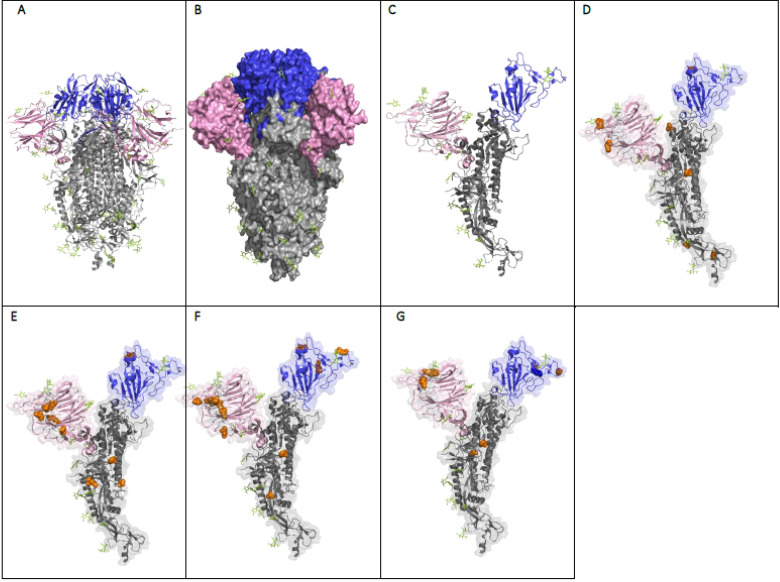
Structural comparison of the spike proteins of the variants of concern and the ancestral variant. (**A**) A ribbon diagram of the prefusion conformation of SARS-CoV-2 S protein trimer (PDB ID: 7DDN) (Ancestral variant). (**B**) SARS-CoV-2 spike protein trimer structure space filling model (B.1) (trimer). (**C**) Ribbon diagram of the monomer of the SARS CoV-2 spike protein (B1.). Spike protein monomer ribbon structures from (**D**) B.1.1.7 (UK variant/Alpha) spike structure in monomer, (**E**) P.1 (Brazil variant/Gamma) spike structure in monomer, (**F**) B.1.357 (South African variant/Beta) spike structure in monomer, and (**G**) B.1.617.2 (Indian variant/Delta) spike structure in monomer. In D to G, the spike protein monomer ribbon structures are superimposed on a faint transparent van der Waals surface The NTD site is shown in pink, the RBD is shown in blue, the mutations in the spike (compared to 19A) are shown as orange spheres, and the glycans are shown in light green. Images were made using PyMOL version 2.3.2.

**Table 1 viruses-13-01911-t001:** List of the current SARS-CoV-2 variants of interest.

WHO Label	Variants	GSAID Lineage	Nextstrain Clade	Date of Designation	Estimated Date of Emergence	Location of Emergence	Mutations (https://covdb.stanford.edu/page/mutation-viewer/#sec_alpha; accessed on 1 July 2021)
Alpha	20I/501Y.V1, VOC 202012/01, or B.1.1.7	GRY (formerly GR/501Y.V1)	20I (V1)	December 2020	September 2020	United Kingdom	ORF1a: T183I (PLpro), A890D (PLpro), I1412T (PLpro), Δ106–108 (nsp6), P323L (RdRP) Spike: Δ69–70, Δ144, N501Y, D614G, P681H, T716I, S982A, D1118H ORF8: insQ27, R52I, Y73C N: D3L, 235F
Beta	20H/501Y.V2 or B.1.351	GH/501Y.V2	20H(V2)	December 2020	May 2020	South Africa	ORF1: T85I (nsp2), K837N (PLpro), H26Y(nsp4), S137L (nsp4), K90R (3CL), Δ106–108 (nsp6), D135Y (RdRP), P323L (RdRP), T588I (nsp3) Spike: L18F, D80A, D215G, Δ242–244, R246I, K417N, E484K, N501Y, D614G, A701V ORF3a: Q57H, 171L E: P71L, N: T205I
Gamma	P.1	GR/501Y.V3	20J (V3)	January 2021	November 2020	Brazil	ORF1: S70L (PLpro), K977Q (PLpro), Δ106–108 (nsp6), P323L (PdRP), E341D (nsp13) Spike: L18F, T20N, P26S, D138Y, R190S, K417T, E484K, N501Y, D614G, H655Y, T1027I, V1176F ORF3a: S253P ORF8: 92K N: P80R, R203K, G204R
Delta	B.1.617.2	G/478K.V1	21A	May 2021(VOI: Apr 2021)	October 2020	India	ORF1: A488S (PLpro), P1228 (PLpro), P1469S (PLpro), V167L (nsp4), T492I (nsp4), T77A (nsp6), P323L (RdRP), G671S (RdRP), P77L (nsp13), A394V (nsp14) Spike: T19R, Δ157–158, L452R, T478K, D614G, P681R, D950N ORF3: S26L M: I82T ORF7a: T120I, V82a ORF7b: T40I ORF8: Δ119–120 N: D63G, R203M, N377Y

**Table 2 viruses-13-01911-t002:** The transmissibly and immune evasion in variants of concern.

VOC	Transmissibility	Immune Evasiveness
Alpha	75% (95% CI: 70–80%) more transmissible than the pre-existing variants from October 2020–November 2020 [57]. Estimated to be 50% more transmissible in Switzerland (report, https://ispmbern.github.io/covid-19/variants/ accessed on 18 June 2021) An estimated 29% increase in transmissibility [58]. 43–90% more transmissible than previous circulating strains. Increased transmissibility of 40–42% across the US [59].	Enhanced innate immune evasion [60]. May be due to the mutations in nsp14 [61]. Mutations in NTD and N501 are associated with virus escape from neutralising antibodies [62,63]. Reduced potency of neutralising antibodies reported. Escape from mainly NTD-specific antibodies [64,65] No significant difference to 4.5-fold reduction in neutralisation of alpha variant and the ancestral strain by convalescent plasma [66,67,68,69,70] No significant difference to a 2.6-fold reduction in neutralisation of alpha variant and the ancestral strain by post vaccination (BNT162b2) sera [66,67,69,71,72]. No significant difference to a 1.8-fold reduction in neutralisation of alpha variant and the ancestral strain by post vaccination (mRNA-1273) sera [66,70,73]. Reduced (8.9 fold) ability of post-vaccination (ChAdOx1) sera to neutralise [74].
Beta	Estimated to be 1.5 times more transmissible than the previous variants [75]. An estimated 25% increase in transmissibility [58].	Resistance to antibody mainly due to RBD mutations (K417N, E484K, N501Y) [76]. Reduced (7.9 to 13.3-fold) neutralisation of beta variant compared with the ancestral strains by convalescent plasma [65,67,68,77,78,79]. Reduced ability (4.9 to 16-fold) of post-vaccination (BNT162b2) sera to neutralise [64,65,67,68,80]. Reduced ability (12.4-fold) of post-vaccination (mRNA-1273) sera to neutralise [65,73]. Reduced (9 fold to undetectable) ability of post-vaccination (ChAdOx1) sera to neutralise [64,68,79,81]. Reduced ability (9.7- and 14.5-fold reduction) of post-vaccination (mRNA-1273 or NVX-CoV2373 respectively) sera to neutralise [82].
Gamma	An estimated 38% increase in transmissibility [58]. Between 1.7 to 2.4 times more transmissible than the earlier circulating variants [83]. Could be more transmissible due to the higher viral load in the upper respiratory tract compared to prior circulating variants [84].	Resistance to antibody mainly due to RBD mutations (K417N, E484K, N501Y) [76] Reduced (3.1 to 8.2-fold) neutralisation of gamma variant compared with the ancestral strains by convalescent plasma [68,85,86]. Reduced ability (2.6 to 5.1-fold) of post-vaccination (BNT162b2) sera to neutralise [68,85,86,87]. Reduced ability (4.8-fold) of post-vaccination (mRNA-1273) sera to neutralise [73,86]. Reduced (2.9-fold to undetectable) ability of post-vaccination (ChAdOx1) sera to neutralise [68,87].
Delta	An estimated 97% increase in transmissibility compared to non-VOCs [58].	Escape due to the combination of RBD mutations (L452R, T478K) and NTD mutations [64,76] Reduced (2.4 to 6-fold) neutralisation of delta variant compared with the ancestral strains by convalescent plasma [64,68,88]. Reduced ability (2.5 to 11.3-fold) of post-vaccination (BNT162b2) sera to neutralise [64,68,72,80,88]. Reduced ability (3-fold) of post-vaccination (mRNA-1273) sera to neutralise [88]. Reduced (4.3 to 5-fold) ability of post-vaccination (ChAdOx1) sera to neutralise [64,68].

## Data Availability

Not applicable.

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
