# Peer review of "The Immune Response to SARS-CoV-2 and Variants of Concern"

_viruses, 2021, doi:10.3390/v13101911_

Round 1
Reviewer 1 Report
This is nice review on the humoral and cellular immune responses to SARS-CoV-2 in the context of mutation and structural/functional changes especially in the emerging variants of concern.
Minor comment
My question is whether the artwork on the Figure 4 is made by the authors or is there a reproduction from another publication, if so, the source should be added before publication.
Author Response
We thank the reviewer for their positive assessment of our manuscript.
Figure 4 was made by us using PyMOL. We have added this information to the figure legend.
Reviewer 2 Report
This review by Torbati and colleagues addressed a very interesting and useful issue since, in my opinion, the variants of concern of SARS-CoV-2 are not sufficiently discussed and explained from the molecular point of view. The virus molecular mechanisms of evasion of the immune responses (humoral and cellular), and how this aspect impacts upon the immune protection are in depth examined, and the manuscript is well structured. Each topic is debated with clarity and in a logical manner, resulting in a high quality and intriguing review paper.
Concerning the theories around the emergence of the new variants, I would suggest to debate the issue of the possibility to mix different vaccines up, in order to improve and make more effective the cellular and humoral immune responses. This is certainly one of the latest interesting aspect in this variable scenario and I think it could enhance the scientific value of the manuscript, making the review more complete.
Author Response
We thank the reviewer for their positive assessment of our manuscript, and in particular for noting that it fills a gap in the literature.
We also thank them for their excellent suggestion that we add a discussion on heterologous prime-boost vaccination strategies as a way of addressing variants of concern. In response, we have added another section entitled "Heterologous prime-boost vaccination and variants of concern" to the end of the manuscript discussing this issue.
We have also added a brief discussion on emerging data from Israel about the early effectiveness of a homologous booster dose of BNT162b2 in >60 year olds. This has been added to the section entitled "The efficacy and effectiveness of the vaccines against variants of concern".
These changes have been coloured in red in the revised manuscript.